

# Phylogenomic analysis of 589 metagenome-assembled genomes encompassing all major prokaryotic lineages from the gut of higher termites

Vincent Hervé[1], Pengfei Liu[1], Carsten Dietrich[1], David Sillam-Dussès[2], Petr Stiblik[3], Jan Šobotník[3] and Andreas Brune[1]

[1] Research Group Insect Gut Microbiology and Symbiosis, Max Planck Institute for Terrestrial Microbiology, Marburg, Germany
[2] Laboratory of Experimental and Comparative Ethology EA 4443, Université Paris 13, Villetaneuse, France
[3] Faculty of Forestry and Wood Sciences, Czech University of Life Sciences, Prague, Czech Republic

Corresponding authors
Vincent Hervé,
vincent.herve8@gmail.com
Andreas Brune,
brune@mpi-marburg.mpg.de

## ABSTRACT

"Higher" termites have been able to colonize all tropical and subtropical regions because of their ability to digest lignocellulose with the aid of their prokaryotic gut microbiota. Over the last decade, numerous studies based on 16S rRNA gene amplicon libraries have largely described both the taxonomy and structure of the prokaryotic communities associated with termite guts. Host diet and microenvironmental conditions have emerged as the main factors structuring the microbial assemblages in the different gut compartments. Additionally, these molecular inventories have revealed the existence of termite-specific clusters that indicate coevolutionary processes in numerous prokaryotic lineages. However, for lack of representative isolates, the functional role of most lineages remains unclear. We reconstructed 589 metagenome-assembled genomes (MAGs) from the different gut compartments of eight higher termite species that encompass 17 prokaryotic phyla. By iteratively building genome trees for each clade, we significantly improved the initial automated assignment, frequently up to the genus level. We recovered MAGs from most of the termite-specific clusters in the radiation of, for example, Planctomycetes, Fibrobacteres, Bacteroidetes, Euryarchaeota, Bathyarchaeota, Spirochaetes, Saccharibacteria, and Firmicutes, which to date contained only few or no representative genomes. Moreover, the MAGs included abundant members of the termite gut microbiota. This dataset represents the largest genomic resource for arthropod-associated microorganisms available to date and contributes substantially to populating the tree of life. More importantly, it provides a backbone for studying the metabolic potential of the termite gut microbiota, including the key members involved in carbon and nitrogen biogeochemical cycles, and important clues that may help cultivating representatives of these understudied clades.

## INTRODUCTION

Termites (Blattodea: Termitoidae) are eusocial insects that have predominantly and successfully colonized tropical and subtropical areas across the world. One of the keys to this success is their rare ability to degrade lignocellulose, a very abundant but recalcitrant complex carbon substrate (*Cragg et al., 2015*). As major decomposers, termites play an important role in carbon cycling (*Yamada et al., 2005*; *Dahlsjö et al., 2014*; *Liu et al., 2015*; *Griffiths et al., 2019*). Lignocellulose digestion by termites is attributed to the presence of a specific microbiota colonizing the different gut compartments of the host (*Brune, 2014*). Even though termites produce endogenous cellulases in the labial glands and/or midgut (*Tokuda et al., 2004*; *Fujita, Miura & Matsumoto, 2008*), the digestive processes in the hindgut are the result of microbial activities.

"Lower" termites feed almost exclusively on wood, whereas "higher" termites (Termitidae family) diversified their diet and extended it from wood to plant litter, humus, and soil (*Donovan, Eggleton & Bignell, 2001*). Higher termites represent the most diverse and taxon-rich clade and form about 85% of the termite generic diversity (*Krishna et al., 2013*). Their gut morphology is more complex than that of the basal clades, and is characterized by the presence of a mixed-segment and an enlarged proctodeal segment P1. Moreover, the gut displays strong variations in pH and oxygen partial pressure along the anterior–posterior axis, which creates microenvironments within the gut (*Brune, 2014*).

Termites harbor a specific and complex gut microbiota (*Brune & Dietrich, 2015*; *Bourguignon et al., 2018*). Over the last decade, numerous studies targeting the 16S rRNA gene have cataloged the prokaryotic diversity of the termite gut microbiota. By analyzing the structure and composition of these microbial communities, the roles of host taxonomy (*Dietrich, Köhler & Brune, 2014*; *Abdul Rahman et al., 2015*), host diet (*Mikaelyan et al., 2015a*), and microenvironments found in the different gut compartments (*Mikaelyan, Meuser & Brune, 2017*) have emerged as the main factors shaping the termite gut microbiota. These studies have also highlighted patterns of dominant taxa associated with specific diet and/or gut compartment (*Mikaelyan, Meuser & Brune, 2017*). For instance, Spirochaetes tend to be the dominant phylum in the gut of wood/grass feeders, whereas their abundance is lower in litter, humus, and soil feeders, in which Firmicutes are much more abundant. The accumulated 16S rRNA gene reads have revealed the existence of termite-specific clusters among both bacterial and archaeal phyla (e.g., among Fibrobacteres, Clostridia, Spirochaetes, and Euryarchaeota).

All these studies focusing on the 16S rRNA gene have helped microbiologists in answering the question "who is there?," but the following questions "what are they doing?" and "who is doing what?" remain open. Attempts to answer the latter questions have been made, for example, by analyzing different fractions of the gut content of *Nasutitermes* spp., which led to the identification of fiber-associated cellulolytic bacterial taxa (*Mikaelyan et al., 2014*), or by focusing on the diversity of individual functional marker genes, such as *nifH* (*Ohkuma, Noda & Kudo, 1999*) or formyl-tetrahydrofolate synthetase (*Ottesen & Leadbetter, 2011*). The latter approach, however, is problematic because the organismal origin of the respective genes is often obfuscated by frequent horizontal gene transfers
**Table 1 Recovery of metagenome-assembled genomes (MAGs) from the 30 termite gut metagenomes analyzed in this study.** The host termite, its mitochondrial genome accession number, dietary preference, and the originating gut compartments are indicated. *C* crop (foregut), *M* midgut, *P1–P5* proctodeal compartments (hindgut). The sample codes used for the metagenomes are the combination of host ID and gut compartment.

| Termite species | ID | Mitogenome | Diet | Number of MAGs | | | | | | |
|---|---|---|---|---|---|---|---|---|---|---|
| | | | | C | M | P1 | P3 | P4 | P5 | Total |
| *Microcerotermes parvus* | Mp193 | KP091690 | Wood | –[a] | – | 1 | 1 | 4 | – | 6 |
| *Nasutitermes corniger* | Nc150 | KP091691 | Wood | 0 | 1 | 3 | 6 | 9 | 1 | 20 |
| *Cornitermes* sp. | Co191 | KP091688 | Litter | – | – | 32 | 22 | 7 | – | 61 |
| *Neocapritermes taracua* | Nt197 | KP091692 | Humus | – | – | 6 | 70 | 11 | – | 87 |
| *Termes hospes* | Th196 | KP091693 | Humus | – | – | 6 | 64 | 27 | – | 97 |
| *Embiratermes neotenicus* | Emb289 | KY436202 | Humus | – | – | 45 | 52 | 21 | – | 118 |
| *Labiotermes labralis* | Lab288 | KY436201 | Soil | – | – | 66 | 72 | 31 | – | 169 |
| *Cubitermes ugandensis* | Cu122 | KP091689 | Soil | 0 | 0 | 5 | 5 | 3 | 18 | 31 |

**Note:**
[a] Not sequenced.

between prokaryotes. Thus, it has been suggested that genome-centric instead of gene-centric approaches are much more relevant for elucidation of soil or gut microbiotas (*Prosser, 2015*). Unfortunately, the number of available isolates of termite gut microbiota and their genomes (*Zheng & Brune, 2015*; *Yuki et al., 2018*) are low compared to those from other environments. However, modern culture-independent methods, namely metagenomics and single-cell genomics have recently allowed the generation of numerous metagenome-assembled genomes (MAGs) and single-amplified genomes (SAGs), respectively, from uncultivated or difficult to cultivate organisms (*Albertsen et al., 2013*; *Woyke, Doud & Schulz, 2017*). MAGs are becoming increasingly more prominent in the literature (*Bowers et al., 2017*) and populate the tree of life (*Parks et al., 2017*). Additionally, MAGs offer the opportunity to explore the metabolic potential of these organisms and to link it with their ecology.

To date, only a limited number of MAGs and SAGs of uncultured bacteria have been recovered from the guts of higher termites; these represent termite-specific lineages of Fibrobacteres (*Abdul Rahman et al., 2016*) and Cyanobacteria (*Utami et al., 2018*). Here, we applied a binning algorithm to 30 metagenomes from different gut compartments of eight higher termite species encompassing different feeding groups to massively recover hundreds of prokaryotic MAGs from these samples. After quality filtering, all these MAGs were taxonomically identified within a phylogenomic framework and are discussed in the context of insect gut microbiology and symbiosis.

# MATERIALS AND METHODS

## Metagenomic datasets

To cover a wide range of microbial diversity, we used 30 metagenomic datasets representing the main gut compartments (crop, midgut, P1–P5 proctodeal compartments of the hindgut) and main feeding groups present in higher-termites (see Table 1).
Eight species of higher termites, identified by both morphological criteria and analysis of the mitogenome, were considered: *Cornitermes* sp., *Cubitermes ugandensis*, *Microcerotermes parvus*, *Nasutitermes corniger*, *Neocapritermes taracua*, *Termes hospes* (*Dietrich & Brune, 2016*), *Labiotermes labralis*, and *Embiratermes neotenicus* (*Hervé & Brune, 2017*). Field experiments were approved by the French Ministry for the Ecological and Solidarity Transition (UID: ABSCH-CNA-FR-240495-2; permit TREL1902817S/118). Processing of the termite samples and DNA extraction and purification were described previously (*Rossmassler et al., 2015*). Metagenomic libraries were prepared, sequenced, quality controlled, and assembled at the Joint Genome Institute (Walnut Creek, CA, USA). DNA was sequenced using Illumina HiSeq 2000 or Illumina HiSeq 2500 (Illumina Inc., San Diego, CA, USA). Quality-controlled reads were assembled and uploaded to the Integrated Microbial Genomes (IMG/M ER) database (*Markowitz et al., 2014*). Accession numbers and information about these 30 metagenomes can be found in Table S1.

## Genome reconstruction

For each metagenomic dataset, both quality-controlled (QC) and assembled (contigs) reads were downloaded from IMG/M ER in August 2017. To obtain coverage profile of contigs from each metagenomic assembly, the QC reads were mapped to contigs using BWA v0.7.15 with the bwa-mem algorithm (*Li & Durbin, 2009*). This generated SAM files that were subsequently converted into BAM files using SAMtools v1.3 (*Li et al., 2009*). Combining coverage profile and tetranucleotide frequency information, genomes were reconstructed from each metagenome with MetaBAT version 2.10.2 with default parameters (*Kang et al., 2019*). Quality of the reconstructed genomes was estimated with CheckM v1.0.8 (*Parks et al., 2015*). Only MAGs that were at least 50% complete and with less than 10% contamination, were retained for subsequent analyses. These MAGs have been deposited at the Sequence Read Archive (SRA) under the BioProject accession number PRJNA560329; genomes are available with accession numbers SRR9983610–SRR9984198 (Table S2). Additionally, the MAGs have been deposited at the NCBI's Assembly Database under the accessions WQRH00000000–WRNX00000000 (Table S2).

For each MAG, CheckM was also used to extract 16S rRNA gene sequences as well as a set of 43 phylogenetically informative marker genes consisting primarily of 29 ribosomal proteins (PF00466, PF03946, PF00298, PF00572, PF00238, PF00252, PF00861, PF00687, PF00237, PF00276, PF00831, PF00297, PF00573, PF00281, PF00673, PF00411, PF00164, PF00312, PF00366, PF00203, PF00318, PF00189, PF03719, PF00333, PF00177, PF00410, PF00380, PF03947, PF00181), nine RNA polymerase domains (PF04563, PF04997, PF00623, PF05000, PF04561, PF04565, PF00562, PF04560, PF01192), two tRNA ligases (TIGR00344 and TIGR00422), a signal peptide binding domain (PF02978), a translation-initiation factor 2 (PF11987) and a TruB family pseudouridylate synthase (PF01509). Finally, CheckM was also used for a preliminary taxonomic classification of the MAGs by phylogenetic placement of the MAGs into the CheckM reference genome tree.

## Phylogenomic analysis

In order to improve the initial CheckM classification, genome trees were built for each clade of interest (from kingdom to family level). Using this initial CheckM classification and when available, the 16S rRNA gene classification, genomes of closely related organisms and relevant outgroups were manually selected and downloaded from NCBI and IMG/M ER. These genomes were subjected to a similar CheckM analysis to extract a set of 43 single-copy marker genes, to translate them into amino acid sequences, and to create a concatenated fasta file (6,988 positions). For each clade of interest, the amino acid sequences from the MAGs, their relatives, and outgroups were aligned with MAFFT v7.305b and the FFT-NS-2 method (*Katoh & Standley, 2013*), and the resulting alignment was filtered using trimAl v1.2rev59 with the gappyout method (*Capella-Gutierrez et al., 2009*). Smart Model Selection (*Lefort, Longueville & Gascuel, 2017*) was used to determine the best model of amino acid evolution of the filtered alignment based on Akaike Information Criterion. Subsequently, a maximum-likelihood phylogenetic tree was built with PhyML 3.0 (*Guindon et al., 2010*). Branch supports were calculated using a Chi2-based parametric approximate likelihood-ratio test (aLRT) (*Anisimova & Gascuel, 2006*). Finally, each tree was visualized and edited with iTOL (*Letunic & Bork, 2019*). Following the procedure described above, a genome tree containing only the MAGs generated in the present study was also built and visualized with GraPhlAn version 0.9.7 (*Asnicar et al., 2015*).

## Placement of MAGs in a 16S rRNA-based phylogenetic framework

All 16S rRNA gene sequences recovered from the respective bins were classified using the phylogenetic framework of the current SILVA reference database (SSURef NR 99 release 132) (*Quast et al., 2013*). The database was manually curated to extend the taxonomic outline of all relevant lineages to genus level by linking the taxonomy to the termite-specific groups to that of the DictDb v3 database (*Mikaelyan et al., 2015b*). 16S rRNA gene sequences contained in the MAGs were aligned with SINA version 1.2.11 (*Pruesse, Peplies & Glöckner, 2012*) and imported into the reference database. Sequences longer than >100 bp were added to the reference trees using the parsimony tool of ARB version 6.0.6 (*Ludwig et al., 2004*). If none of the MAGs in a cluster contained a 16S rRNA gene longer than 100 bp, or if the placement of the 16S rRNA genes in the bin conflicted with the results of the phylogenomic analysis (indicating a contamination), the phylogenomic classification was used.

## Estimation of the relative abundance of the MAGs in each metagenome

For each metagenome, raw reads were mapped against MAGs using BWA (*Li & Durbin, 2009*) with default parameters. Unmapped reads and reads mapped to more than one location were removed by using SAMtools (*Li et al., 2009*) with parameters: F 0x904. Reads mapped to each MAGs were summarized using the "pileup.sh" script (BBmap 38.26) (*Bushnell, 2014*). The relative abundance of each MAG was calculated as the total number of reads mapped to a MAG divided by the total number of reads in the corresponding metagenome sample, as described in *Hua et al. (2019)*. Similarly, the MAG coverage was

estimated by multiplying the mapped reads by the read length and dividing it by the MAG length.

## Statistical analyses

Statistical analyses were performed with R version 3.4.4 (*R Development Core Team, 2019*), and data were visualized with the *ggplot2* (*Wickham, 2016*) package. Correlations between quantitative variables were investigated with Pearson's product moment correlation coefficient.

# RESULTS AND DISCUSSION

## Metagenomes and MAGs overview

Metagenomic reads were generated from the P1, P3, and P4 proctodeal compartments of the gut of the two termite species *E. neotenicus* and *L. labralis*. These six metagenomes were combined with 24 previously published metagenomes from the gut of higher termites (*Rossmassler et al., 2015*) in order to obtain data encompassing different gut compartments from eight species of higher termites feeding on different lignocellulosic substrates ranging from wood to soil (Table 1). Metagenomic binning of these 30 termite gut metagenomes yielded 1,732 bins in total (Table S1). For further analysis, we selected only those bins that represented high-quality (135 bins, >90% complete, and <5% contamination) and medium-quality (454 bins, >50% complete, and <10% contamination) MAGs (Table 1; Table S1). The present study focused on these 589 MAGs, which showed on average a 38.6-fold coverage (Table S2).

The number of MAGs recovered from the different metagenomes did not show a Gaussian distribution. Instead, we found a significant and positive relationship between the number of metagenome-assembled reads and the number of MAGs recovered ($r = 0.85$, $p < 0.0001$), indicating that assembly success and sequencing depth were important predictors of genome reconstruction success (Fig. 1). This is in agreement with benchmarking reports on metagenomic datasets (*Sczyrba et al., 2017*) and underscore that a good quality assembly is a prerequisite for high binning recovery, which is important to consider when designing a metagenomic project for the purpose of binning. A significantly higher number of assembled reads and of MAGs recovered was observed in the current dataset compared to the *Rossmassler et al. (2015)* dataset (Wilcoxon test, $p < 0.005$), highlighting the importance of this new dataset (Fig. 1).

## MAGs taxonomy and abundance

We investigated the phylogenomic context of the 589 MAGs. An initial automated classification of the MAGs using CheckM and when available, the taxonomic assignment of the 16S rRNA gene, identified representatives of 15 prokaryotic phyla (Table S3). Initially, 142 MAGs (24% of the dataset) remained unclassified at the phylum level, and key taxa of the termite microbiota, such as Fibrobacteres and *Treponema*, were absent or only poorly represented. This is partly explained by the lack of representative genomes for certain taxa in the reference genome tree provided in the current version of CheckM (e.g., only one Fibrobacteres genome and one Elusimicrobia genome, and an

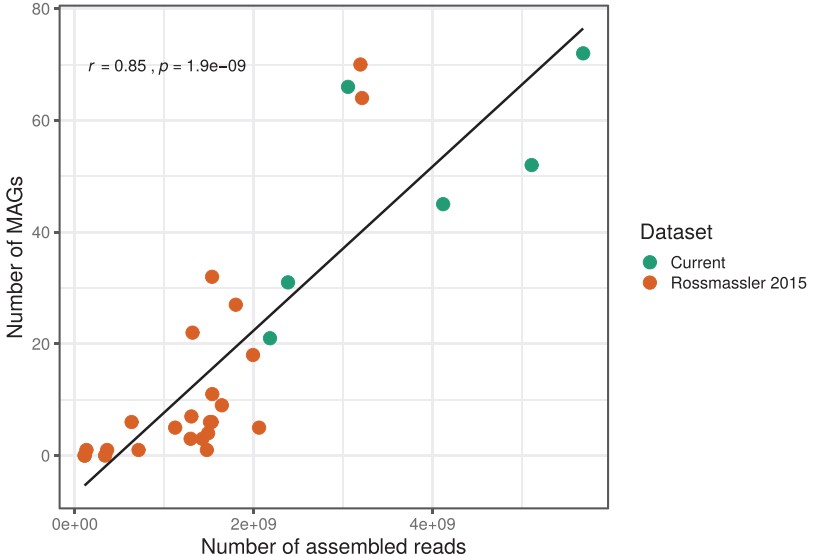

**Figure 1** **Relationship between the number of MAGs recovered and the number of assembled reads in the respective metagenomes.** The linear regression line and the Pearson correlation coefficient ($r$) are shown for the entire dataset.

absence of Bathyarchaeota and Kiritimatiellaeota genomes). New tools incorporating larger databases, such as GTDB-Tk (*Parks et al., 2018*; *Chaumeil et al., 2019*), will probably resolve such issues.

We improved the taxonomic resolution of the classification by iteratively constructing genome trees for each clade of interest that included all recently published reference genomes. This approach allowed the successful classification of all 589 MAGs, in some cases down to the genus level (Table S2). Thirty-eight MAGs were from the archaeal domain, and 551 MAGs were from the bacterial domain, which together represented a total of 17 prokaryotic phyla (Fig. 2). The taxonomic diversity of MAGs recovered is broadly representative of that observed in previously published 16S rRNA surveys, suggesting good taxonomic coverage of termite-associated prokaryotes from the different gut compartments and host diets (Figs. S1 and S2).

The MAG taxonomy was further refined by placing all 16S rRNA genes recovered from the bins into the phylogenetic framework of the current SILVA reference database, which allowed classifying most of the MAGs down to genus level (Table S2). When we compared the taxa represented by the MAGs to the distribution of the corresponding taxa in amplicon libraries of the bacterial gut microbiota of a representative selection of higher termites that were classified using the same framework (*Lampert, Mikaelyan & Brune, 2019*), we found a high level of congruence between the datasets. The MAGs represented 15 of the 19 bacterial phyla in the amplicon libraries that comprised >0.1% of all reads, including all core phyla (represented in >80% of all host species) with the exception of Verrucomicrobia (Fig. 3). A high representation in the amplicon libraries of the taxa represented by MAGs was confirmed at all taxonomic ranks down to the genus level

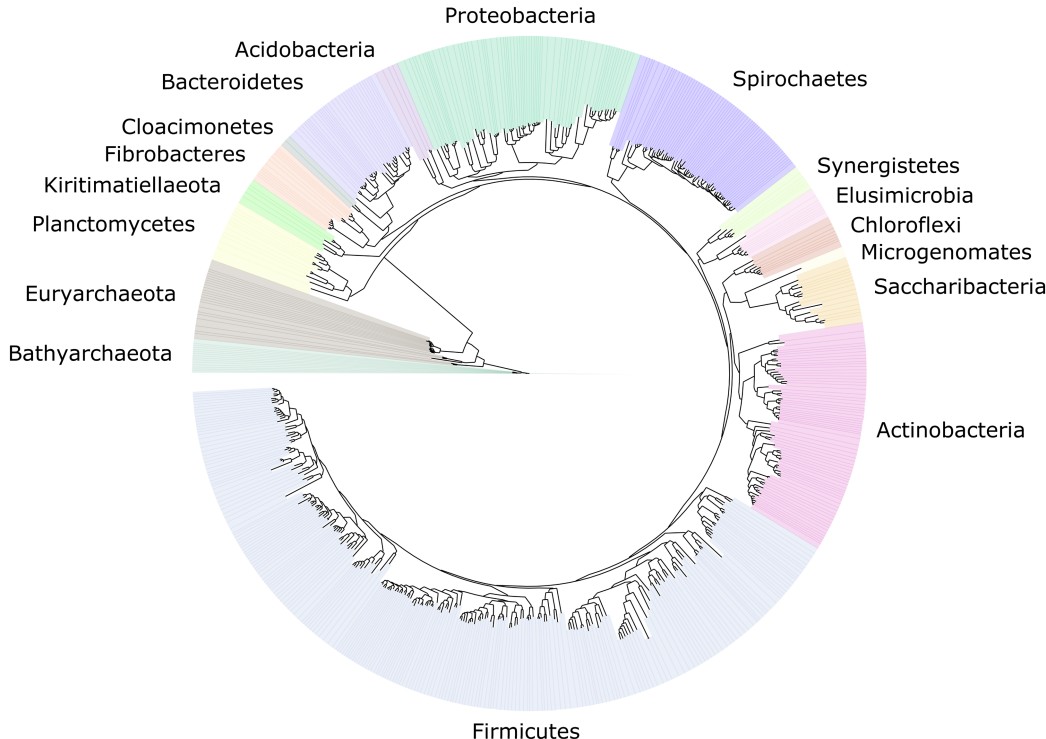

**Figure 2 Distribution of the 589 MAGs among bacterial and archaeal phyla.** This maximum-likelihood tree was inferred from a concatenated alignment (amino acids) of 43 protein-coding genes (6,801 positions) using the LG+G+I model of evolution.

(Table S4), underscoring that the present dataset covers the majority of lineages that colonize the higher-termite gut.

We computed the relative abundance of each MAG. These abundances ranged from 0.005% to 4.03% (Table S2), with a mean value of 0.19%. These values indicated that the present dataset includes major taxonomic groups of the termite gut microbiota, which was confirmed when we looked at the taxonomic distribution of the MAGs. Considering the MAG relative abundance and not only their presence within samples, we could observe an effect of the host diet on the taxonomic distribution (Fig. 4). Indeed, similarities were observed when we compared taxonomic patterns of the MAG relative abundance with previously published 16S rRNA gene amplicon-based surveys (*Abdul Rahman et al., 2015*; *Mikaelyan, Meuser & Brune, 2017*). For instance, Spirochaetes were the most abundant phylum within the wood-feeding termite *N. corniger*, and their proportion decreases along the humification gradient, being less abundant in the gut of humus feeders and litter feeders and even less abundant in soil feeders, in the favor of other phyla such as Firmicutes. Fibrobacteres were preferentially abundant within wood- and litter-feeder samples (Fig. 4). Interestingly, a significant and negative relationship between the number of metagenome-assembled reads in a sample and the MAG relative abundances within this sample ($r = -0.33$, $p < 0.0001$) was observed across all the samples. This could be partly explained by the fact that increasing sequencing depth would increase the number of metagenome-assembled reads and thus allow the

| Phylum | MAGs | | | Core | 16S rRNA amplicon libraries Average abundance (%) | | | | | |
|---|---|---|---|---|---|---|---|---|---|---|
| | Total number | with 16S gene | Rel. abd. (%) | | All | Wood | Litter | Humus | Soil | Fungus |
| Acidobacteria | 4 | 2 | 0.4 | • | 0.5 | 0.7 | 0.4 | 0.8 | 0.4 | 0.2 |
| Actinobacteria | 71 | 22 | 3.0 | • | 3.1 | 2.0 | 1.8 | 2.1 | 5.2 | 3.9 |
| Bacteroidetes | 33 | 5 | 2.7 | • | 17.5 | 7.6 | 22.1 | 18.5 | 12.1 | 28.7 |
| Chloroflexi | 8 | 5 | 0.1 | | 0.2 | 0.0 | 0.1 | 0.3 | 0.8 | 0.0 |
| Cloacimonetes | 2 | 2 | 0.2 | | 0.4 | 0.0 | 0.2 | 1.6 | 0.2 | 0.0 |
| Deferribacteres | – | – | | | 0.3 | 0.0 | 0.0 | 0.1 | 0.1 | 1.1 |
| Elusimicrobia | 9 | 2 | 0.4 | • | 0.8 | 0.1 | 1.4 | 1.9 | 0.3 | 0.5 |
| Epsilonproteobacteria | – | – | | | 1.5 | 0.3 | 0.7 | 0.0 | 0.2 | 6.1 |
| Fibrobacteres | 13 | 2 | 6.6 | • | 3.2 | 8.5 | 6.8 | 0.6 | 0.1 | 1.1 |
| Firmicutes | 237 | 73 | 9.2 | • | 33.5 | 13.1 | 26.0 | 41.0 | 56.2 | 28.7 |
| Fusobacteria | – | – | | | 0.2 | 0.1 | 0.0 | 0.0 | 0.1 | 0.7 |
| Kiritimatiellaeota | 5 | 3 | 0.3 | | 0.3 | 0.1 | 0.2 | 0.3 | 0.9 | 0.1 |
| Microgenomates | 1 | 1 | 0.6 | • | 0.6 | 0.1 | 0.1 | 0.2 | 0.4 | 2.3 |
| Planctomycetes | 12 | 8 | 0.7 | | 1.0 | 0.0 | 0.1 | 0.0 | 0.6 | 3.7 |
| Proteobacteria | 67 | 23 | 9.8 | • | 7.8 | 3.5 | 8.0 | 7.7 | 9.4 | 10.6 |
| Saccharibacteria | 15 | 12 | 0.1 | • | 2.2 | 0.4 | 3.1 | 1.9 | 0.8 | 5.2 |
| Spirochaetes | 68 | 10 | 3.0 | • | 24.5 | 62.1 | 27.5 | 21.1 | 8.1 | 4.7 |
| Synergistetes | 6 | 1 | 0.3 | • | 1.5 | 0.8 | 0.5 | 1.2 | 3.2 | 1.5 |
| Verrucomicrobia | – | – | | | 0.4 | 0.1 | 0.3 | 0.2 | 0.6 | 0.6 |
| **Bacteria** | **551** | **170** | **36.7** | | **100** | **100** | **100** | **100** | **100** | **100** |
| Bathyarchaeota | 15 | 10 | 1.2 | | | | | | | |
| Euryarchaeota | 23 | 6 | 0.5 | | | | | | | |
| **Archaea** | **38** | **16** | **1.7** | | | | | | | |
| **Total** | **589** | **187** | **38.4** | | | | | | | |

**Figure 3 Phylum-level representation of MAGs among the bacterial gut microbiota of higher termites.** The average abundance of the corresponding lineages in 16S rRNA amplicon libraries of higher termites from different diet groups is shown for comparison. Core lineages represented in at least 80% of these samples are marked. For an interactive spreadsheet resolving each lineage to genus level, see Supplemental Information (Table S4).

binning of sequences from less abundant organisms. However, since quantity of metagenome-assembled reads and relative abundance are not independent variables, it also implies that MAG relative abundances can not be directly quantitatively compared between samples but only within a single sample. Thus, proportions of taxa within a sample using relative abundance can be used to describe such sample.

## Archaea

The archaeal domain was represented by members of the phyla Euryarchaeota and Bathyarchaeota (Fig. 5; Fig. S3). Euryarchaeota were represented by 23 MAGs that were classified as members of the genera *Methanobrevibacter* (family *Methanobacteriaceae*; three MAGs) and, *Methanimicrococcus* (family *Methanosarcinaceae*; three MAGs), and members of the family *Methanomethylophilaceae* (16 MAGs), nine of them in the genus *Candidatus* Methanoplasma. MAGs assigned as Euryarchaeota encompassed three (*Methanobacteriales*, *Methanosarcinales*, and *Methanomassiliicoccales*) of the four orders of methanogens found in termite guts (*Brune, 2019*); *Methanomicrobiales* were absent from the present dataset. This genomic resource will be extremely valuable for a better understanding of the genomic basis of methanogenesis in the termite gut and more generally for investigating the functional role of archaea in arthropod guts. Indeed, Euryarchaota have been found to be present in virtually all termite species investigated (*Brune, 2019*), and a global 16S rRNA gene survey has revealed that this phylum is the most abundant archaeal clade in arthropod-associated microbiota (*Schloss et al., 2016*). Bathyarchaeota were represented by 15 MAGs, which formed a termite-specific cluster,

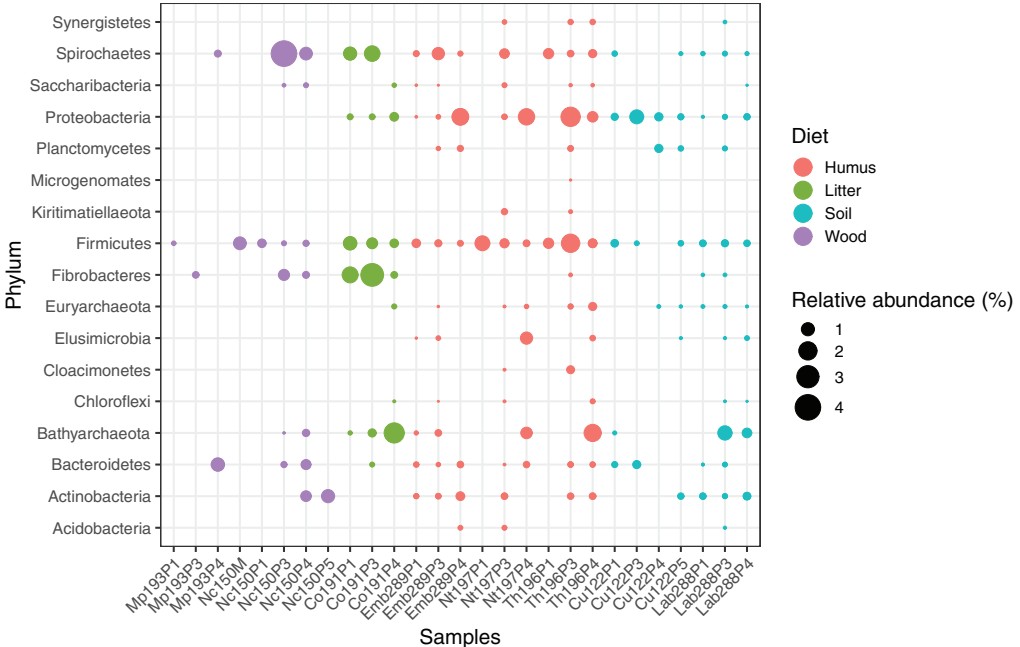

**Figure 4** **Relative abundance of the MAGs from different phyla among the respective metagenomes.** Circle size indicates the relative abundance of the MAGs among the respective metagenome sample; color indicates host diet. To estimate the relative abundance of each MAG, the total number of reads mapped to a MAG was divided by the total number of reads in the corresponding metagenome sample.

with Bathyarchaeota reconstructed from sediments of the White Oak River (WOR) estuary (NC, USA) as next relatives (*Lazar et al., 2016*) (Fig. 5). Bathyarchaeota is a lineage formerly known as Miscellaneous Crenarchaeota Group (MCG), which has been reported to occur in the gut of soil-feeding termites (*Friedrich et al., 2001*). To date, MAGs of Bathyarchaeota have been mostly derived from aquatic environments (*Zhou et al., 2018*). Here, we identify the members of the termite gut lineage as Bathyarchaeota and provide the first genomes from this environment. Interestingly, Bathyarchaeota MAGs were particularly abundant in humus-, litter-, and soil-feeding termites (Fig. 4); a genomic characterization, combined with analyses of their abundance and localization, should shed light on the metabolic potential of these organisms and their functional role in termite guts.

## Firmicutes

Firmicutes was by far the phylum with the highest number of MAGs, but also the phylum with the highest average relative abundance (33.5%) in 16S rRNA gene-based surveys (Fig. 3). The 237 MAGs (40% of the total dataset) represented three classes (*Bacilli*, *Clostridia*, and *Erysipelotrichia*) and ten families, including four members of *Streptococcaceae* (*Bacilli*) and three members of *Turicibacteraceae* (*Erysipelotrichia*). *Clostridia* was the most diverse and rich class (229 MAGs), in which *Ruminococcaceae* (95 MAGs), *Defluviitaleaceae* (67 MAGs), *Lachnospiraceae* (four MAGs), *Peptococcaceae*

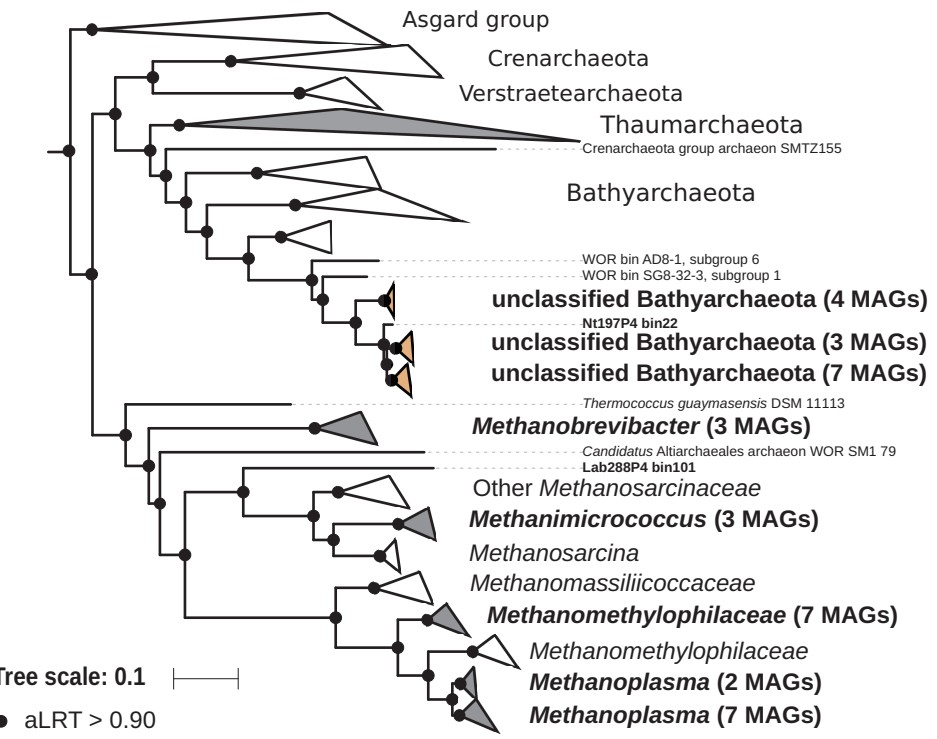

**Figure 5 Phylogenomic tree of the archaeal domain.** This maximum-likelihood tree was inferred from a concatenated alignment of 43 proteins (6,682 positions) using the LG+G+I+F model of amino-acid evolution. Branch supports were calculated using a Chi2-based parametric approximate likelihood-ratio test. Names in bold included MAGs recovered in the present study. Clusters shaded in brown consist exclusively of MAGs from termite guts and clusters shaded in gray contain genomes from termite guts. The Asgard group was used as outgroup.

(16 MAGs), *Christensenellaceae* (nine MAGs), *Eubacteriaceae* (two MAGs), Family XIII *incertae sedis* (six MAGs), and Clostridiales vadinBB60 group (22 MAGs) families were identified. These high numbers of *Ruminococcaceae* and *Defluviitaleaceae* MAGs were reflected by high relative abundances of these two families in 16S rRNA gene-based surveys (15.9% and 3.1% for the *Ruminococcaceae* and *Defluviitaleaceae*, respectively; Table S4). Interestingly, among the *Defluviitaleaceae*, the genomes were mainly recovered from the P1 compartment (53 MAGs, i.e., 79% of the family members) whereas *Ruminococcaceae* were predominantly recovered from the P3 compartment (59 MAGs, i.e., 62% of the family members). Further studies should investigate the potential metabolic specialization of these two families in relation to the gut physicochemical properties. A fourth class-level lineage could not be further classified for lack of reference genomes. In a recent global 16S rRNA gene-based survey, it has been suggested that many novel lineages of Firmicutes in insect-associated metagenomes are hidden (*Schulz et al., 2017*). Our present study confirms this idea but our genome trees also provide evidence of new lineages. Here, we report the first genomes of uncultured termite-specific lineages (Table S4) that were already detected in previous 16S rRNA gene-based surveys (*Bourguignon et al., 2018*). For example, the phylogenomic tree of the most abundant

family *Ruminococcaceae* (Fig. S4) showed various termite-specific clusters, including a cluster of 18 MAGs closely related to *Sporobacter termitidis* isolated from *Nasutitermes lujae* (*Grech-Mora et al., 1996*). *Lachnospiraceae*, *Ruminococcaceae*, *Turicibacteraceae* (previously classified as *Erysipelotrichaceae*), and *Defluviitaleaceae* (previously classified as *Lachnospiraceae*) have been reported among the dominant taxa in termite guts (*Mikaelyan, Meuser & Brune, 2017*), but most of them remain uncultivated and/or with few representative genomes. As such, many questions regarding their ecology and metabolism remain open. With 237 Firmicutes MAGs recovered from different gut compartments and from hosts with different diets, the present study provides the material for further genomic exploration of the role of these bacteria in plant polysaccharide degradation, based for instance on CAZyme distribution (*Lombard et al., 2014*). Since diet has been shown to be the main factor shaping gut community composition in higher termites (*Mikaelyan et al., 2015a*), one might hypothesize the existence of different arsenals of lignocellulolytic enzymes, potentially reflecting the host diet specificity (balance between cellulose, lignin, and hemicelluloses). More generally, Firmicutes and especially *Ruminococcaceae* are also abundant and metabolically important in rumen systems (*Svartström et al., 2017*; *Söllinger et al., 2018*; *Stewart et al., 2018*). At a broader scale, our dataset will allow comparative studies between intestinal tract microbiota of ruminants and phytophagous or xylophagous invertebrates, which would allow a better understanding of plant polysaccharide degradation across the tree of life.

## Actinobacteria

Actinobacteria was the second most abundant phylum with 71 MAGs, including members of the classes *Acidimicrobiia*, *Actinobacteriia*, *Coriobacteriia*, and *Thermoleophilia* (Fig. S5). Eleven families were represented, namely *Propionibacteriaceae* (12 MAGs), *Promicromonosporaceae* (three MAGs), Clostridiales *incertae sedis* (16 MAGs), OPB41 (16 MAGs) *Cellulomonadaceae* (seven MAGs), *Frankiaceae* (one MAG), *Sanguibacteraceae* (four MAGs), *Microbacteriaceae* (two MAGs), *Nocardioidaceae* (two MAGs), *Acidimicrobiaceae* (one MAG), *Nocardiaceae* (one MAG), and *Conexibacteraceae* (one MAG). Among these 71 MAGs, 36 were recovered from humus feeders, 33 from soil feeders but only two from wood feeders, which suggests a higher prevalence in termites with a more humified diet. This phylum is known to be present and of significant abundance in both the nest (*Sujada, Sungthong & Lumyong, 2014*) and gut of termites (*Le Roes-Hill, Rohland & Burton, 2011*), but to be more abundant in the nest (*Moreira et al., 2018*). This was for instance the case for the families *Acidimicrobiaceae*, *Nocardiaceae*, *Promicromonosporaceae*, *Microbacteriaceae*, *Nocardioidaceae*, and *Propionibacteriaceae*, which were more abundant in the nest than in the gut of workers or soldiers of *Procornitermes araujoi* (*Moreira et al., 2018*). Therefore, one of the key questions regarding this phylum concerns their effective role in lignocellulose degradation in the termite guts. Are they just present in the surrounding environment of the termite and thus sometimes transit from the gut or are they actively involved in food digestion? The MAGs obtained in the present study will allow to address such questions by

evaluating gene expression of these organisms using metatranscriptomic data from higher termites (*He et al., 2013*; *Marynowska et al., 2017*).

## Spirochaetes

The phylum Spirochaetes was represented by 68 MAGs from wood-, soil-, litter-, and humus-feeding termites. It has long been known that Spirochaetes are a diverse and important lineage in termite gut (*Paster et al., 1996*; *Lilburn, Schmidt & Breznak, 1999*), especially because of their involvement in reductive acetogenesis (*Leadbetter et al., 1999*; *Ohkuma et al., 2015*) and in hemicellulose degradation (*Tokuda et al., 2018*). In terms of abundance, Spirochaetes are among the dominant phyla in termite guts (Fig. 3) and may represent more than half of the bacterial relative abundance in some species (*Diouf et al., 2018a*). Three Spirochaetes orders, namely *Brevinematales* (one MAG), *Leptospirales* (four MAGs) and *Spirochaetales* (59 MAGs), were identified (Fig. 6; Fig. S6). In the latter order, 54 MAGs recovered from the P1, P3, and P4 compartments of wood-, litter-, humus-, and soil-feeding hosts were assigned to the termite-specific cluster *Treponema* I (*Ohkuma, Iida & Kudo, 1999*; *Lilburn, Schmidt & Breznak, 1999*) and represent the first genomes of this cluster from higher termites. Indeed, to date only two *Treponema* I genomes are available, and both were recovered from isolates, namely *Treponema azotonutricium* and *Treponema primitia*, from the hindgut of the lower termite *Zootermopsis angusticollis* (*Graber, Leadbetter & Breznak, 2004*). Thus, our dataset significantly expands the genomic resources for this taxonomic group. Subclusters of this clade have been identified on a dedicated genome tree (Fig. 6). The genome tree topology is in agreement with a previous phylogenomic Spirochaetes study (*Gupta, Mahmood & Adeolu, 2013*). Regarding Spirochaetes classification, our tree topology suggests that the genus *Treponema* could be elevated at least to the family rank due to the presence of distinct *Treponema* clusters (Fig. 6). This observation is also in agreement with the recent Genome Taxonomy Database, which now proposes a *Treponemataceae* family and a *Treponematales* order (*Parks et al., 2018*; *Chaumeil et al., 2019*).

## Fibrobacteres

Members of the phylum Fibrobacteres are abundant in the hindgut of wood-feeding higher termites (Fig. 3) (*Hongoh et al., 2006*), where they have been identified as fiber-associated cellulolytic bacteria (*Mikaelyan et al., 2014*). Here, 13 members of the Fibrobacteres phylum were recovered from the P1, P3, and P4 compartments of wood-, litter-, humus-, and soil-feeding termites. These genomes encompass the three orders, namely *Chitinispirillales* (*Sorokin et al., 2016*), previously known as TG3 subphylum 1 (*Hongoh et al., 2006*, five MAGs), *Chitinivibrionales* (previously known as TG3 subphylum 2; two MAGs), and *Fibrobacterales* (six MAGs). While a previous study of termites guts had already provided MAGs of *Chitinivibrionaceae* and *Fibrobacteraceae* and documented their fiber-degrading capacities (*Abdul Rahman et al., 2016*), the present study provides the five first genomes of the termite-associated members of *Chitinispirillaceae* (Fig. 7).

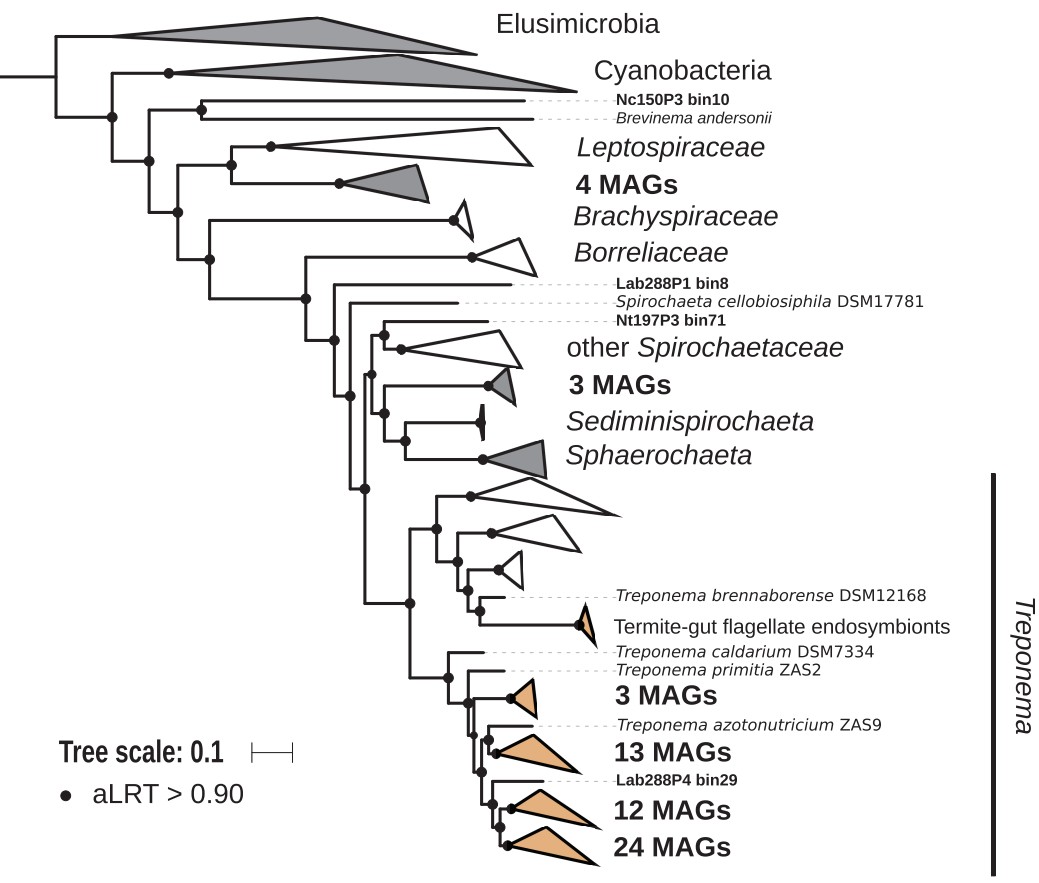

**Figure 6 Phylogenomic tree of the Spirochaetes phylum.** This maximum-likelihood tree was inferred from a concatenated alignment of 43 proteins (6,741 positions) using the LG+G+I+F model of amino-acid evolution. Branch supports were calculated using a Chi2-based parametric approximate likelihood-ratio test. Names in bold included MAGs recovered in the present study. Clusters shaded in brown consist exclusively of genomes from termite guts and clusters shaded in gray contain genomes from termite guts. Elusimicrobia and Cyanobacteria were used as outgroup.

Phylogenomic analysis indicates that the MAGs classified as Fibrobacterales represent a termite-specific cluster among *Fibrobacteraceae* that comprises *Candidatus* Fibromonas termitidis and forms a sister group to the genus *Fibrobacter* (Fig. 7; Fig. S7). This is in agreement with a previous study that identified the same lineage (but classified as family *Fibromonadaceae*) by 16S rRNA gene-based and phylogenomic analyses (*Abdul Rahman et al., 2016*). None of the MAGs fell into the genus *Fibrobacter*, which was absent also in all 16S rRNA gene-based surveys of termite gut microbiota (*Hongoh et al., 2006*; *Mikaelyan et al., 2015b*; *Bourguignon et al., 2018*). Members of this genus have been isolated from the gastrointestinal tracts of mammals and bird herbivores (*Neumann, McCormick & Suen, 2017*), where they are potentially involved in cellulose and hemicellulose degradation (*Neumann & Suen, 2018*). This suggests co-evolutionary patterns among different Fibrobacteres clades within animal hosts with a lignocellulose-based diet.

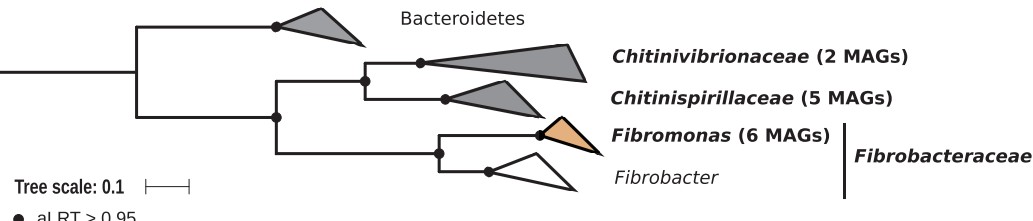

**Figure 7 Phylogenomic tree of the Fibrobacteres phylum.** This maximum-likelihood tree was inferred from a concatenated alignment of 43 proteins (6,516 positions) using the LG+G+I+F model of amino-acid evolution. Branch supports were calculated using a Chi2-based parametric approximate likelihood-ratio test. Names in bold included MAGs recovered in the present study. Clusters shaded in brown consist exclusively of genomes from termite guts and clusters shaded in gray contain genomes from termite guts. Bacteroidetes were used as outgroup.

## Proteobacteria and Bacteroidetes

Sixty-seven MAGs of Proteobacteria belonging to *Alphaproteobacteria* (23 MAGs), *Gammaproteobacteria* (20 MAGs), and *Deltaproteobacteria* (24 MAGs) were recovered from all hindgut compartments of litter-, humus-, and soil-feeding termites. Among the *Deltaproteobacteria*, seven orders were identified, namely *Desulfobacterales* (four MAGs, all assigned to *Desulfobulbus*), *Desulfovibrionales* (five MAGs, all *Desulfovibrionaceae*), *Desulfuromonadales* (one MAG), *Myxococcales* (five *Myxococcaceae* and four *Polyangiaceae*), *Adiutricales* (one MAG), *Syntrophobacterales* (one MAG), and MBNT15 group (two MAGs). *Desulfovibrionaceae* (*Desulfovibrionales*) members of gut and termite-gut clusters have been found to be highly prevalent in termite guts (*Bourguignon et al., 2018*). Similarly, we identified three *Desulfovibrionaceae* MAGs that form a monophyletic clade and two *Desulfovibrionaceae* MAGs that fall into a cluster of gut-associated genomes (Fig. S8). This family, among others, is composed of various sulfate-reducing bacteria; this functional group has already been identified in different termite species (*Kuhnigk et al., 1996*). Thus, these MAGs could provide new genomic resources to further investigate this metabolism in the termite gut.

Our dataset comprises 33 MAGs of Bacteroidetes (Fig. S9), including members of the families Cluster V (four MAGs), RC9 gut group (six MAGs), *Paludibacteraceae* (two MAGs, both assigned to the *Paludibacter* genus), *Rikenellaceae* (nine MAGs), *Marinilabiliaceae* (one MAG), and *Prolixibacteraceae* (one MAG). These Bacteroidetes were found in the P1, P3, and P4 compartments and in wood-, litter-, humus-, and soil-feeding termites. In Blattodea guts, different clusters of *Alistipes* (Bacteroidetes) have been found in a 16S rRNA gene survey (*Mikaelyan et al., 2015b*). Two MAGs from *L. labralis* belonging to the *Rikenellaceae* family and closely related to *Alistipes* have been identified. Additionally, among Bacteroidetes, four MAGs, all originating from P4 compartments, fall into the Cluster V family that contains symbionts of flagellates from guts of lower termites (*Hongoh et al., 2008b*; *Yuki et al., 2015*). We also recovered two MAGs assigned to *Paludibacter*; *Paludibacter propionicigenes* and *Paludibacter jiangxiensis* are both strictly anaerobic, propionate-producing bacteria isolated from rice paddy field (*Ueki et al., 2006*; *Qiu et al., 2014*). Propionate is produced by fermenting bacteria in the

gut of termites (*Odelson & Breznak, 1983*); these bacteria utilize glucose generated by cellulose degradation to form succinate and propionate (*Tokuda et al., 2014*). *P. propionicigenes* might be involved in nitrogen fixation, as *nifH* transcripts assigned to this species are the most abundant in the gut of the wood-feeding beetle *Odontotaenius disjunctus* (*Ceja-Navarro et al., 2014*).

## Saccharibacteria, Synergistetes and Planctomycetes

Fifteen MAGs of *Candidatus* Saccharibacteria (also known as candidate division TM7) were reconstructed from the P1, P3, and P4 compartments of wood-, litter-, humus-, and soil-feeding termites (Fig. S10). Most of them originated from humus feeders (11 MAGs), especially from the P3 compartment (eight out of these 11 MAGs). Similarly, six MAGs of Synergistetes, all belonging to the *Synergistaceae* family that contains a termite/cockroach cluster (*Mikaelyan et al., 2015b*), were recovered from the P3 and P4 compartments of humus- and soil-feeding termites (Fig. S11). Both Saccharibacteria and Synergistetes were recently highlighted as numerically important clades of the termite gut microbiota, with some OTUs being present in the gut of the majority of 94 termite species collected across four continents (*Bourguignon et al., 2018*). They were also contributing to the core microbiota of higher termites (Fig. 3). Genomic analysis of these MAGs should help in understanding the roles of these bacteria in termite gut and also provide clues for designing successful isolation media to study their physiology.

Twelve MAGs were assigned to the phylum Planctomycetes, including four to the class *Phycisphaerae* (and among them two classified as *Tepidisphaerales* CPla-3 termite group), one to class vadinHA49 and seven to the class *Planctomycetia* (all classified as *Pirellulaceae*) (Fig. S12). These MAGs were recovered from the P3, P4, and P5 compartments and were restricted to humus- and soil-feeding termites. The recovery of Planctomycetes was expected, especially from the *Pirellulaceae* family, which also contains termite/cockroach clusters (*Mikaelyan et al., 2015b*). Interestingly, we found three MAGs from the P4 and P5 compartments of *C. ugandensis*, with one 16S rRNA gene sequence assigned to the Rs-B01 termite group, described in a previous study investigating the gut microbiota of the same termite species (*Köhler et al., 2008*). When such 16S rRNA gene information is available, it will allow the direct linkage between prokaryotic taxonomy and potential metabolisms.

## Other phyla

Nine members of the phylum Elusimicrobia were identified, including members of the class *Endomicrobia* (eight members) and *Elusimicrobia* (one member) (Fig. S13). These were found in all hindgut compartments and were restricted to humus- and soil-feeding termites. Currently, only three complete genomes of Elusimicrobia from insect guts are available: *Elusimicrobium minutum* from the gut of a humivorous scarab beetle larva (*Herlemann et al., 2009*), and *Endomicrobium proavitum* (*Zheng & Brune, 2015*) and *Candidatus* Endomicrobium trichonymphae (*Hongoh et al., 2008a*) from the termite gut. Here, we provided nine additional genomes from the guts of humus- and soil-feeding termites.

The Chloroflexi phylum was represented by eight MAGs (all *Dehalococcoidia)*, including seven belonging to the family *Dehalococcoidaceae* and one to the family *Dehalobiaceae*, found exclusively in the P3 and P4 compartments of humus- and soil-feeding termites (Fig. S10). Their function in termite gut remains unclear, but Chloroflexi, including *Dehalococcoidia*, were found to be enriched in lignin-amended tropical forest soil (*DeAngelis et al., 2011*), where oxygen concentration and redox potential are highly variable, as in the termite gut (*Brune, 2014*). Therefore, their ability to use oxygen as final electron acceptor and their potential involvement in lignin degradation could be investigated by comparative genomics.

Minor phyla were also present in our dataset. Two MAGs assigned as Cloacimonetes (Fig. S14) and five MAGs assigned as Kiritimatiellaeota were recovered from the P3 compartment of the two humus-feeding termites *N. taracua* and *T. hospes* (Fig. S15). Kiritimatiellaeota have been reported to be present in the digestive tract of various animals (*Spring et al., 2016*). The few clones obtained from termite guts, which had been tentatively classified as uncultured Verrucomicrobia, were mostly obtained with planctomycete-specific primers (*Köhler et al., 2008*), underscoring the potential biases in amplicon-based studies toward certain taxa. Similarly, one MAG of Microgenomates (also known as candidate division OP11), which probably represents a lineage of Pacebacteria that was discovered only in a recent amplicon-based analysis but occurs in the majority of termites investigated (*Bourguignon et al., 2018*), was reconstructed from the P3 compartment of *T. hospes* (Fig. S10).

Finally, four MAGs classified as Acidobacteria were reconstructed from either the P3 or P4 compartments of humus- and soil-feeding termites (Fig. S16), which show a moderately alkaline or circumneutral pH in comparison to the highly alkaline P1. Of these four genomes, two were assigned to the M1PL1-36 termite group within the family *Holophagaceae* and one to the *Acidobacteriaceae* family. Acidobacteria can represent a significant fraction of the termite gut microbiota, especially in wood-feeding termites (*Hongoh et al., 2005*; *Wang et al., 2016*; *Bourguignon et al., 2018*). In the gut of higher termites, this phylum is present in the core microbiota (Fig. 3). Moreover, *Holophagaceae* and *Acidobacteriaceae* have been reported to be present in moderately acidic lignocellulosic substrates, such as peatland soil (*Schmidt et al., 2015*) and decaying wood (*Hervé et al., 2014*). Genomic analysis should help us in identifying the metabolic potential of these MAGs for lignocellulose degradation.

## Phyla not represented by MAGs

Several bacterial phyla and one archaeal phylum containing prominent taxa that have been identified in previous 16S rRNA gene surveys of termite guts were not represented among the MAGs recovered in the present study. They include Cyanobacteria (class *Melainabacteria*; *Utami et al., 2018*), Lentisphaerae (*Köhler et al., 2012*; *Sabree & Moran, 2014*), Verrucomicrobia (*Wertz et al., 2012*), and Thaumarchaeota (*Friedrich et al., 2001*; *Shi et al., 2015*). Also intracellular symbionts of termite tissues, such as *Wolbachia* (Proteobacteria) (*Diouf et al., 2018b*) were not recovered. Possible reasons are a low relative abundance and/or a high phylogenetic diversity of the respective lineages.

Although larger metagenomes should improve the chances of their recovery in the medium- and high-quality bins, targeted single-cell based approaches have proven to be quite effective in recovering these genomes (*Ohkuma et al., 2015*; *Yuki et al., 2015*; *Utami et al., 2019*).

## CONCLUSIONS

The 589 MAGs reported here represent the largest genomic resource for arthropod-associated microorganisms available to date. We recovered representatives of almost all major prokaryotic lineages previously identified in 16S rRNA gene amplicon-based surveys of the gut of higher termites from the metagenomes. This provides the foundations for studying the metabolism of the prokaryotic gut microbiota of higher termites, including the key members involved in carbon and nitrogen biogeochemical cycles, and important clues that may help in cultivating representatives of these understudied clades.

## ACKNOWLEDGEMENTS

The authors thank all JGI staff, particularly their project manager Tijana Glavina del Rio, for their excellent service. The technical assistance of Katja Meuser is highly appreciated. We also thank the three anonymous reviewers and the editor for their helpful and constructive comments.

### Funding

This study was funded by the Deutsche Forschungsgemeinschaft in the collaborative research center SFB 987 (Microbial Diversity in Environmental Signal Response) and by the Max-Planck-Gesellschaft. The work conducted by the U.S. Department of Energy Joint Genome Institute, a DOE Office of Science User Facility, is supported by the Office of Science of the U.S. Department of Energy under Contract No. DE-AC02-05CH11231. Petr Stiblik and Jan Šobotník were supported by grant "EVA4.0," No. CZ.02.1.01/0.0/0.0/16_019/0000803 financed by OP RDE. The funders had no role in study design, data collection and analysis, decision to publish, or preparation of the manuscript.

### Grant Disclosures

The following grant information was disclosed by the authors:
Deutsche Forschungsgemeinschaft in the collaborative research center SFB 987 (Microbial Diversity in Environmental Signal Response).
Max-Planck-Gesellschaft.
U.S. Department of Energy Joint Genome Institute, a DOE Office of Science User Facility.
Office of Science of the U.S. Department of Energy: DE-AC02-05CH11231.
OP RDE "EVA4.0": CZ.02.1.01/0.0/0.0/16_019/0000803.

### Competing Interests

The authors declare that they have no competing interests.

## Author Contributions

- Vincent Hervé conceived and designed the experiments, performed the experiments, analyzed the data, prepared figures and/or tables, authored or reviewed drafts of the paper, and approved the final draft.
- Pengfei Liu performed the experiments, analyzed the data, authored or reviewed drafts of the paper, and approved the final draft.
- Carsten Dietrich conceived and designed the experimentes, performed the experiments, authored or reviewed drafts of the paper, and approved the final draft.
- David Sillam-Dussès performed the experiments, authored or reviewed drafts of the paper, and approved the final draft.
- Petr Stiblik performed the experiments, authored or reviewed drafts of the paper, and approved the final draft.
- Jan Šobotník performed the experiments, authored or reviewed drafts of the paper, and approved the final draft.
- Andreas Brune conceived and designed the experiments, performed the experiments, analyzed the data, prepared figures and/or tables, authored or reviewed drafts of the paper, and approved the final draft.

## Field Study Permissions

The following information was supplied relating to field study approvals (i.e., approving body and any reference numbers):

Field experiments were approved by the French Ministry for the Ecological and Solidarity Transition (UID: ABSCH-CNA-FR-240495-2) (TREL1902817S/118).

## Data Availability

The data are available at BioProject: PRJNA560329. Genomes are available at: SRR9983610–SRR9984198. The MAGs are available at the NCBI's Assembly Database: WQRH00000000–WRNX00000000 (Table S2).

## Supplemental Information

Supplemental information for this article can be found online at http://dx.doi.org/10.7717/peerj.8614#supplemental-information.

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
