# Peer review of "Phylogenomic analysis of 589 metagenome-assembled genomes encompassing all major prokaryotic lineages from the gut of higher termites"

_PeerJ, doi:10.7717/peerj.8614_

## Round 0.1 · original submission · Minor Revisions

While I do not agree with referee 3 that the manuscript is not a publishable unit, I hope you will take the comments raised by this referee in earnest. Please respond to all concerns raised by the reviewers and in particular:

- Confusion over the information present on L254 concerning the correlation between metagenome-assembled reads and MAG relative abundance. Can you please also clarify how MAG relative abundance was calculated? It appears it is the proportion of reads mapped to a MAG divided by the number of reads mapped to assembled contig. Would it not be more informative and accurate to take the proportion of reads mapped to a MAG divided by the total number of reads in the metagenome (regardless of whether they map to a contig)?

- Referee 3’s suggestion to add a table indicating the presence or absence of key marker genes for different metabolisms. I would be interested to know if the Bathyarchaeota MAGs contain genes comprising the MCR complex. Other readers are likely to have similar questions regarding the metabolic potential of these MAGs.

All 3 referees indicated that the MAGs recovered as part of this study are of high value to the community and a primary contribution of this study. As such, I request that you submit your MAGs to an INSDC genome assembly database (e.g., NCBI’s Assembly database) in addition to the Sequence Read Archive. The majority of bioinformatic tools and databases retrieve genomes from NCBI’s Assembly database and are unlikely to include genomes that only reside in the SRA.

I look forward to reading your revised manuscript.

Regards,
Donovan Parks

Reviewer 1 ·

Basic reporting

The paper is well written and concise with good-shaped (supplementary) tables and figures. English text is excellent. I just list some minor suggestions and questions:

1. Please italicized phylum names (e.g., Fibrobacteres, Proteobacteria...) as the authors did for other taxonomic ranks (e.g., class Clostridia, family Lachnospiraceae...).

2. L233-234: It is difficult to follow why “genomes were recovered among the most abundant phyla from different gut compartments and diets” indicated "a good coverage of the diversity among gut compartments and host diets". Please explain more.

3. L236: "...which can be considered as abundant (Delmont et al. 2018)."
In my opinion, what is "abundant" depends on the diversity of the sample. Anyway, I think that the authors can remove this description. One does not need to declare "0.23" is "abundant" or not.

4. L238: "..abundant members..."
Here, "members" is ambiguous. Readers might regard it means "species". For example, "The present dataset includes major taxonomic groups of termite gut microbiota..." seems better.

5. L247-251: "...a significant and negative relationships between the number of metagenome-assembled reads and the MAG relative abundance..."
Did authors mean here the relationships between the total number of metagenome-assembled reads in a sample and the mean MAG relative abundance of the sample?

6. L364-365: the class Chitinispirillia corresponds to TG3 subphylum 1 and Chitinivibrionia to TG3 sybphylum 2 (please see Sorodokin et al 2016 Frontiers Microbiol).

Experimental design

This paper describes the outline of comparative metagenomic analyses including binning up to 589 high to medium quality prokaryotic genomes (MAGs). This study with such abundant MAGs should be the basis for further detailed studies of termite gut microbiota, which is represented mostly by uncultured lineages. The experimental and analytical procedures follow standard methodologies and enough described.

Validity of the findings

The descriptions are based on their data adequately analyzed and previous findings. No problems are found in "Results and Discussion".

Additional comments

I am looking forward to reading further analyses on these MAGs.

Reviewer 2 ·

Basic reporting

This manuscript is the first to do massive reconstruction of a large number (589) og MAGs from termite guts, spanning termites with different feeding habits. The authors significantly improved the initial automated assignment, frequently up to the genus level, recovering MAGs from most of the termite-specific clusters in the radiation of, e.g., Planctomycetes,
Fibrobacteres, Bacteroidetes, Euryarchaeota, Bathyarchaeota, Spirochaetes,
Saccharibacteria, and Firmicutes, which to date contained only few or no representative
genomes. The dataset is an important genomic resource and contributes substantially to populating the tree of life for gut-associated microbes. It furthermore will likely become an important backbone for a range of different follow-up studies that have been limited by the lack of genomic knowledge on termite bacterial gut members.

The structure, writing and figures in the paper are good, the referencing sufficient, and the language excellent.

Experimental design

The research questions are clear, the methods appropriate (and even extending beyond what many current studies with similar aims do) and the approaches and findings are important for research.

Validity of the findings

The presentation of the results and discussion of the paper is good and thorough, focusing on the more descriptive angles and less on the evolutionary histories that led to the observed patterns, nor their implications for termite-gut bacteria evolution. However, this makes the study solid, void of too much speculation, and very useful as a genomic resource paper, which is a major aim for the authors.

Reviewer 3 ·

Basic reporting

Basic reporting is largely acceptable. The methods section provides a list of methods used and the results are succinctly described.

In general, the primary deficiency in the manuscript as written is that I found the methods and figure legends to be incomplete. The methods and figure legends provide only the most basic information on what was done, without sufficient information regarding how the methods were applied to thoroughly evaluate their results. The most concerning absence is the lack of information regarding how the various genome trees were generated, discussed below in the experimental design section.

However, a more abstract concern is that I’m not completely convinced that this is an adequate publishable unit. I realize that PeerJ does not have a novelty or impact requirement, but in general, I found myself disappointed by the analyses presented. The authors present a series of phylogenetic trees of MAGs, but don’t really do much analysis of any of the MAGS beyond that. Most of the manuscript text is a series of paragraphs summarizing each tree and then writing out a few sentences reviewing what is known about each phylum’s role in the termite. I’m guessing the authors are hoping in the future to then analyze the genomes and use them to gain information about their roles in the termite gut, but without that they’re left with a somewhat empty paper. Even the trees are so summarized as to be largely useless except in the context of future analyses of these groups. Essentially, this manuscript reads as a glorified genome announcement.

I’d really like to see *something* more, even if that’s just pairing their genome trees with 16S trees placing as many of their MAGs as possibly within the phylogenetic framework they laid out in DictDB, and potentially even linking them to 16S-based surveys of gut microbiome composition and providing quantitative analysis of which groups are under or over-represented in MAGs vs 16S. This would at least build on the strength of the manuscript as presented.

Even better would be picking out a couple of pathways/metabolisms of interest and doing a BLAST search for key marker genes of each in all of their genomes. A comprehensive analysis is likely beyond the scope, but even just a list of which groups have well-characterized marker genes for nitrogen fixation, acetogenesis and methanogenesis would help. I realize that the genomes are incomplete, so lack of a marker is not evidence of absence, but presence of key marker genes is still evidence of the capability to carry out that metabolism and could provide new information regarding the roles of these microbes in termite gut metabolism.

Line 230-232. This statement would appear to directly contradict the following paragraph, where you talk about impacts of host diet on taxonomic patterns.

Lines 247-254: I’m having trouble deciphering this section. First, I found it out of place at the end of a paragraph discussing the relative abundances of different bacterial groups across samples. Further, I’m having trouble following exactly what they are trying to say. “The number of assembled reads is negatively correlated with MAG relative abundance”: do they mean the fraction of total reads mapped to all MAGs, the fraction of reads mapped to MAGs that were in medium/high quality MAGs, or the average abundance of medium and high quality MAGs? Why is this useful to know? I can’t make heads or tails of the last two sentences, either, but this may be because I can’t figure out exactly which measures of relative abundance they are referring to in which parts of these sentences.

Experimental design

The largest problem with the experimental design is the lack of information regarding the genome alignments and genome tree construction. The authors specify how the alignments and trees were constructed in terms of program parameters, but don’t lay out the results of those analyses, only the trees. As the genome trees are the primary data being presented, the authors need to provide detailed information regarding their alignments, including, at a minimum, a list of the 43 genes used in their concatenated alignments, how many were present in each of the MAGS, and how the authors compensated for fact that many of the MAGS likely lacked multiple genes. The authors state that they used an automated filtering algorithm for deciding which columns of the alignment were used; the number of alignment columns used in each analysis should be listed in the figure legends, as well as information regarding whether this filter effectively eliminated any of the conserved genes from consideration in construction of the genome trees.

Lines 179-181: It seems odd to me that the authors report ‘relative abundance’ as fraction of mapped reads. As the MAGs differ in length and completeness, shouldn’t a length-corrected (or completeness-corrected) abundance value be provided? Or was this an attempt to simplify? Regardless, the way in which abundance was calculated should be summarized in the Figure 3 legend to make the work more readable.

Validity of the findings

My primary concern is that I can’t really put a finger on what, exactly, their findings are. Even their conclusions section reads mostly as a genome announcement, saying that they have constructed a resource that will be of use to the field as a whole. As far as the rest go, they created a bunch of MAGS and genome trees for those MAGS, but don’t really go much further than that. For all the corresponding author’s interest in termite gut symbiont phylogeny, they don’t even place their MAGS within the phylogenetic framework laid out in DictDB.

On the other hand, at least this publication will call attention to a potentially useful resource for other scientists, which can then be cited in future studies that make use of this data. So I leave it up to PeerJ to determine what a minimally publishable unit is, should the authors be opposed to further revision

Additional comments

There are no obvious scientific errors in what is presented, although as specified above some experimental details are lacking. Further, this was clearly a lot of work. However, given the extent of their dataset, I find myself deeply disappointed that the authors chose not to take their results any further. As is, this is a moderately useful genome announcement that can be cited in future studies using these resources.

---

## Round 0.2 · accepted · Accept

I recommend addressing the issue raised regarding clarity on lines 247 to 252 once you have received the proofing PDF.

Reviewer 3 ·

Basic reporting

Good. The language is largely clear and the article is well-written.

Experimental design

Good. I remain disappointed that the authors do not do more with their data, but agree that this will be a good supplemental resource for future analyses of these genomes. I appreciate the addition of the 16S analysis, and fell it enhances the manuscript.

Validity of the findings

The authors appear to have attempted to clarify lines 247-252: “Obvious patterns in the taxonomic distribution of the MAGs according to the sample origin were not apparent…” This seems to have been an attempt by the authors to clarify a sentence that was previously noted as confusing, but I think they’ve just made it even more confusing, especially as there are clear differences in the recovery of different phyla from different samples (most obviously, only Firmicutes were obtained from the midgut). Maybe just go with “The taxonomic diversity of MAGs recovered is broadly representative of that observed in previously published 16S rRNA surveys, suggesting good taxonomic coverage of termite-associated microbes” or something along those lines?

Additional comments

I appreciate that the genomes are uploaded to the Assembly database, although if Genbank permits it would be helpful to have the annotations uploaded as well rather than simply the raw sequences.